# HyperSPNs:
# Compact and Expressive Probabilistic Circuits

**Andy Shih**
Stanford University
andyshih@cs.stanford.edu

**Dorsa Sadigh**
Stanford University
dorsa@cs.stanford.edu

**Stefano Ermon**
Stanford University
ermon@cs.stanford.edu

## Abstract

Probabilistic circuits (PCs) are a family of generative models which allows for the computation of exact likelihoods and marginals of its probability distributions. PCs are both expressive and tractable, and serve as popular choices for discrete density estimation tasks. However, large PCs are susceptible to overfitting, and only a few regularization strategies (e.g., dropout, weight-decay) have been explored. We propose HyperSPNs: a new paradigm of generating the mixture weights of large PCs using a small-scale neural network. Our framework can be viewed as a soft weight-sharing strategy, which combines the greater expressiveness of large models with the better generalization and memory-footprint properties of small models. We show the merits of our regularization strategy on two state-of-the-art PC families introduced in recent literature – RAT-SPNs and EiNETs – and demonstrate generalization improvements in both models on a suite of density estimation benchmarks in both discrete and continuous domains.

## 1  Introduction

One of the core motivations for building models of probability distributions from data is to *reason* about the data distribution. For example, given a model of the distribution over driving routes, we may like to reason about the probability that a driver will take a certain route given that there is congestion on a certain street. Or, given a distribution over weather conditions, we may like to reason about the probability of it raining $k$ times in the next year. Traditional probabilistic models such as Categorical/Gaussian Mixture Models or Hidden Markov Models (HMMs) are well-equipped to answer these types of queries with exact probabilities that are consistent with the model distribution [1]. However, their learned model distribution might not accurately approximate the data distribution due to their simplicity.

On the other end of the spectrum, the modern wave of deep generative models has largely focused on learning accurate approximations of the true data distribution, at the cost of tractability. High capacity models such as autoregressive or flow models can mimic the true data distribution with great fidelity [33, 13, 22]. However, they are not designed for reasoning about the probability distribution beyond computation of likelihoods. More illustrative of this trend are GANs and EBMs, which are expressive enough to sample high-quality images, but give up even the ability to compute exact likelihoods [9, 14].

To maintain the ability to reason about the distribution induced by our model, we need to explore within tractable probabilistic models. Of these model families, probabilistic circuits (e.g. Sum Product Networks) are one of the most expressive and general, subsuming shallow mixture models, and HMMs while maintaining their tractable properties [6, 26, 34]. Their expressive power comes from their hierarchical / deep architecture, allowing them to express a large number of modes in their distribution. Their tractability comes from global constraints imposed in their network structure, enabling efficient and exact computation of likelihoods, marginals, and more. Probabilistic circuits

35th Conference on Neural Information Processing Systems (NeurIPS 2021).

(PC) are a popular choice for density estimation [35, 18] and approximate inference in discrete settings [30].

Given the status of probabilistic circuits as one of the most expressive models among tractable probabilistic model families, recent works have looked into pushing the limits of expressivity of probabilistic circuits [15, 25, 24]. Naturally, the bigger and deeper the probabilistic circuit, the greater the expressiveness of the model. This has led to a trend of building large probabilistic circuits via design choices such as tensorizing model parameters, forgoing structure learning, and more. For example, [24] report the training of probabilistic circuits with $9.4$M parameters.

However, larger probabilistic circuits are more susceptible to overfitting. In addition, the so-called double-descent phenomenon [19] (wherein highly overparameterized models exhibit low generalization gap) has yet to be observed in probabilistic circuits. This puts even more importance on effective regularization strategies for training large probabilistic circuits. Recent works have used dropout [25] and weight decay [37], but there has not been much exploration of other regularization strategies.

In this work, we propose HyperSPNs, which regularize by aggressively limiting the degrees of freedom of the model. Drawing inspiration from HyperNetworks [11], we generate the weights of the probabilistic circuit via a small-scale external neural network. More precisely, we partition the parameters (mixture weights) of the PC into several sectors. For each sector, we learn a low-dimensional embedding, then map that to the parameters of the sector using the neural network. The generated parameters still structurally form a PC, so we retain the same ability to reason about the probability distribution induced by the HyperSPN.

HyperSPNs combine the greater expressiveness of large probabilistic circuits with the better generalization of models with fewer degrees of freedom. The external neural network has much fewer parameters than the original PC, effectively regularizing the PC through a soft weight-sharing strategy. As a bonus, the memory requirement for storing the model is much smaller, and the memory requirement for evaluating the model can also be drastically reduced by materializing sectors of the PC "on the fly". We verify the empirical performance of HyperSPNs on density estimation tasks for the Twenty Datasets benchmark [12], the Amazon Baby Registries benchmark [8], and the Street View House Numbers (SVHN) [20] dataset, showing generalization improvements over competing regularization strategies in all three settings. Our results suggest that HyperSPNs are a promising framework for training probabilistic circuits.

## 2 Background

Probabilistic circuits (PCs) are a family of generative models known for their tractability properties [6, 26, 34]. They support efficient and exact inference routines for computing likelihoods and marginals/conditionals of their probability distribution. The internals of a PC consist of a network (edge-weighted Directed Acyclic Graph) of sum and product nodes, stacked in a deep/hierarchical manner and evaluated from leaves to root. PCs gain their tractability by enforcing various structural constraints on their network, such as decomposability, which restricts children of the same product node to have disjoint scopes [5, 6]. Different combination of constraints leading to different types of PCs, including Arithmetic Circuits [6], and most notably Sum Product Networks (SPNs) [26].

The probability of an input $\boldsymbol{x}$ on the PC is defined to be the value $g_{\text{root}}(\boldsymbol{x})$ obtained at the root of the PC after evaluating the network topologically bottom-up. Internally, edges leading into a sum node are weighted with mixture probabilities $\alpha$ (that sum to 1), and edges leading into a product node are unweighted. Leaf nodes at the base of the network contain leaf distributions $l$. The nodes of the PC are evaluated as follows, with $ch(i)$ denoting the children of node $i$:

$$g_i(\boldsymbol{x}) = \begin{cases} l_i(\boldsymbol{x}) & i \text{ is leaf node} \\ \sum_{j \in ch(i)} \alpha_{ij} g_j(\boldsymbol{x}) & i \text{ is sum node} \\ \prod_{j \in ch(i)} g_j(\boldsymbol{x}) & i \text{ is product node} \end{cases}$$

As long as the leaf distributions are tractable and the network structure satisfies the constraints of decomposability and smoothness, the output at $g_{\text{root}}$ is guaranteed to be normalized, and corresponding marginals/conditionals can be evaluated with linear-time exact inference algorithms [5, 6].

The network structure of a PC refers to the set of nodes and edge connections in the DAG, excluding the edge weights. There are various ways of constructing the network structure of PCs, e.g., through

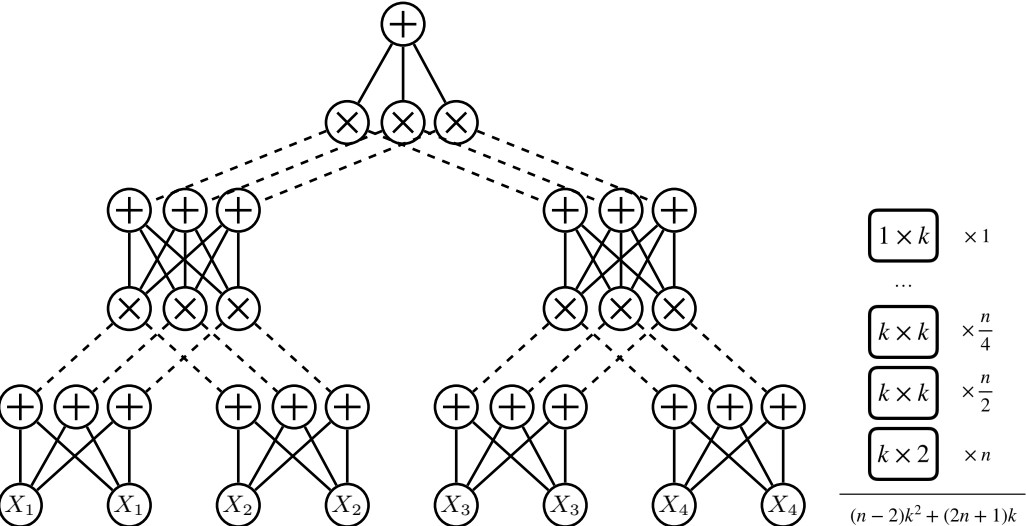

(a) We show the structure of a RAT-SPN over $n = 4$ variables using layer size $k = 3$. Solid edges are weighted with trainable parameters, and dashed edges are unweighted. Each fully connected layer of solid edges can be treated as a sector.

(b) Counting the # of parameters, given the number of variables $n$ and layer size $k$.

Figure 1: Given a RAT-SPN, we can partition its set of parameters into sectors. The trainable parameters (solid edges) in Figure 1a can be grouped into fully-connected layers of size $1 \times k$, $k \times k$, and $k \times 2$, oriented in a binary-tree structure. In Figure 1b, we visualize the sectors as blocks, counting the number of blocks in each layer of the binary tree to arrive at the total parameter count. The sector abstraction is a key ingredient in implementing the HyperSPN's soft weight-sharing strategy.

the use of domain knowledge or through search-based methods [6, 17, 7, 28]. Recently, there has been a trend of abstracting away individual sum and product nodes, and instead dealing directly with layers that satisfy the structural constraints necessary for tractability [25, 24, 30]. Prescribing a network structure through a stacking of these layers leads to memory and hardware efficiencies that allow us to scale up the training of PCs to millions of edges.

Given a PC with a fixed network structure, we can learn its edge weights to best fit the probability distribution of data. We will refer to the full set of mixture probabilities leading into sum nodes in the PC as its parameters. These parameters can be learned using methods such as Expectation-Maximization (EM) [23], signomial program formulations [38], or simply via (projected) gradient descent. EM has been a well-studied choice for optimizing PCs [23, 24], although it requires hand-derived updates that can be tedious and sometimes even incorrect [26]. In this work, we rely on gradient descent, which is straightforward and works out-of-the-box on new families of models [37].

Learning a PC with a large number of parameters can be prone to overfitting. One regularization strategy that has been proposed is "probabilistic dropout" [25], which takes inspiration from the standard dropout technique in neural networks [31] and randomly ignores a subset of edges leading into sum nodes. Weight decay has also been commonly used as another form of regularization [37]. Unfortunately, there has been little exploration done around regularization strategies, and some current approaches only work in limited settings (e.g., dropout has been limited to settings that use PCs as a discriminative classifier [25]).

## 2.1 RAT-SPNs

Two particular PC structures have been recently proposed, namely RAT-SPNs [25] and EiNETs [24], both of which scale to millions of parameters. They are conceptually similar, the main difference being that EiNETs are designed for better GPU performance. To ground our discussion around regularizing large PCs, we will describe our methods just on RAT-SPNs, although we also apply our methods on EiNETs in later experiments too. In RAT-SPNs, the input variables are organized into a binary tree, with one leaf in the binary tree corresponding to each variable. We start from the base

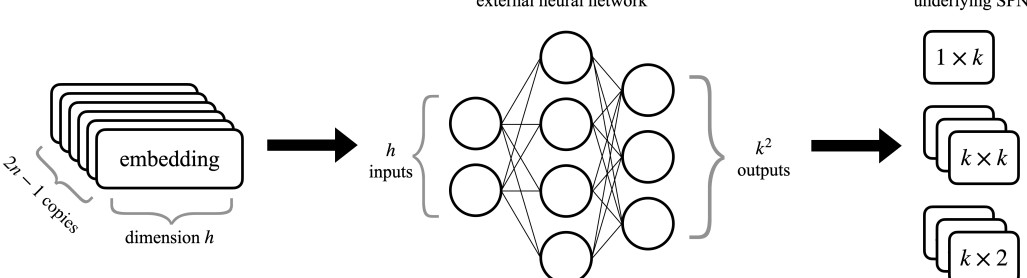

Figure 2: HyperSPN: we introduce a small-scale external neural network that generates the parameters of the underlying SPN. Using the sector abstraction from Figure 1, we learn an embedding of dimension $h$ for each of the $2n-1$ sectors. The neural network maps each embedding to the parameters of its corresponding sector, which materializes the underlying SPN.

of the SPN and construct its full structure by stacking sum layers and product layers. For each leaf node in the binary tree, we create one fully connected sum layer on top of the SPN leaves. For each internal node in the binary tree, we create one product layer that merges its two children layers in the SPN, and then stack one fully connected sum layer on top. We illustrate this construction in Figure 1. Finally, RAT-SPNs create so-called replicas of this structure, by repeating the construction but with permuted orderings of input variables in the binary tree leaves. The replicas are then combined under one top-level mixture distribution (for readability, Figure 1 does not show any replicas).

The size of each sum layer in the RAT-SPN is determined by a single constant $k$. For convenience, we also let the size of each product layer be $k$. This fully specifies the structure of our SPN. For the rest of the paper, it is helpful to abstractly group the parameters of the RAT-SPN into disjoint sectors. Corresponding to the root of the binary tree, we have a sector with $1 \times k$ parameters. Corresponding to each internal node in the binary tree, we have a sector with $k \times k$ parameters. Finally, corresponding to each leaf in the binary tree, we have a sector with $k \times l$ parameters, where $l$ is the number of SPN leaf distributions per variable (we use $l = 2$ unless otherwise stated).

As shown in Figure 1, for $n$ variables (where $n$ is a power of 2) and $r$ replicas, this corresponds to $2n-1$ sectors totaling $O(rnk^2)$ parameters. In particular, for each replica there are $(n-2)k^2 + (2n+1)k$ parameters. For example, in a problem domain with 1000 variable, using layer size $k = 10$ and replica $r = 10$ leads to a RAT-SPN of around 1M parameters.

## 3 HyperSPNs

We propose HyperSPNs, an approach of regularizing probabilistic circuits with a large number of parameters by limiting the degrees of freedom of the model using an external neural network. We describe our method concretely on RAT-SPNs, which we will refer to as simply SPNs.

To regularize the SPN, we introduce a smaller-scale external neural network to generate the parameters of the SPN (see Figure 2). We associate each of the sectors of the SPN with a sector embedding. For each sector, we map the embedding to the parameters of the sector via the neural network. Then, we use these mapped parameters directly as the parameters of the SPN, re-normalize such that parameters leading into a sum node add up to 1, and evaluate the SPN as usual. We visualize the computation graph of this process in Figure 3. We materialize the SPN in the forward pass, and propagate

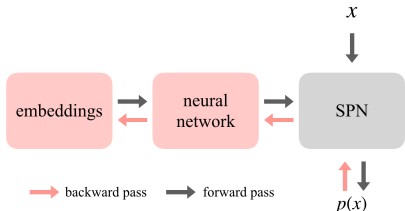

Figure 3: We depict the computation graph when training a HyperSPN. The red modules have trainable parameters, whereas the gray module does not. Red arrows show the flow of gradients during backpropagation.

the gradients through the SPN in the backward pass. We optimize the neural network and embeddings jointly using gradient descent.

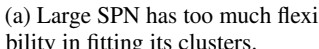

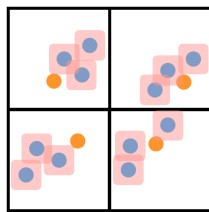 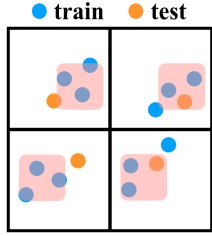 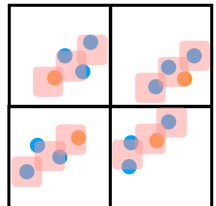

(a) Large SPN has too much flexibility in fitting its clusters.

(b) Small SPN is limited in the flexibility and number of clusters.

(c) HyperSPN has many clusters, with shared degree of freedom.

Figure 4: We visualize the soft weight-sharing effect in HyperSPNs, with training data in blue and testing data in orange. HyperSPNs use a large underlying SPN structure, allowing for a large number of clusters (3 in each quadrant) to fit the data well. Since the SPN parameters correspond to directly mixture weights, the weight-sharing strategy in HyperSPNs prevent overfitting by limiting the degrees of freedom across clusters. We depict this as fixing the shape of the cluster-triples across quadrants.

HyperSPNs can aggressively reduce the degrees of freedom of the original SPN. Using embeddings of size $h$, the number of learnable parameters is reduced to $O((rn + k^2)h)$. For the same above example with $n = 1000, k = 10, r = 10$, choosing embeddings of dimension $h = 5$ reduces the number of learnable parameters 20 fold, from around 1M down to around 50k parameters.

The lower degrees of freedom mitigates overfitting of the model. At the same time, the materialized SPN has a large number of mixture weights, and thus can express distribution families with a greater number of clusters compared to a small SPN with the same degrees of freedom. We visualize this with a motivating example in Figure 4, where we represent a dataset using a grid that is split recursively for each variable (we only draw the splits for the first 2 variables; the rest is implied). Intuitively, the number of parameters of the SPN is tied closely to the number of clusters in the distribution. As a result, training a large SPN can produce many clusters that overfit to the training points in blue (Figure 4a), and training a small SPN can lead to coarse factorizations and inflexible clusters (Figure 4b). Both cases can lead to poor generalization on the testing points in orange.

On the other hand, the HyperSPN uses a soft weight-sharing strategy by storing its parameters as low-dimensional embeddings and decoding them through a small-scale shared neural network. This allows the model to learn a high number of clusters without suffering from too high of a degree of freedom. In Figure 4c, we visualize the soft weight-sharing as fixing the shape of the cluster-triples within each quadrant, while still giving the model control over the center position of each cluster-triple. In this way, HyperSPNs can improve generalization compared to a large SPN with the same number of mixture weights and a small SPN with the same degrees of freedom. We verify this phenomenon on a hand-crafted dataset in the Appendix, and further observe similar improvements on real world datasets in our experiments.

## 3.1 Memory-Efficient Evaluation

Another merit of HyperSPNs is the ability for low-memory storage of the model, since we only need to store the embeddings and the neural network, which together have much fewer parameters than the SPN. Importantly, our abstraction of the SPN structure into sectors allows us to forgo storage of the nodes and edges in the SPN network. Moreover, not only can we be memory-efficient during storage , but we can also be memory-efficient during evaluation. To do so, we materialize sectors of the SPN on the fly, and erase them from memory when they become unnecessary.

The sectors of the SPN are structured as a binary tree, so to evaluate each sector we need all the input values leading into the sector. These input values correspond to the output values (each a vector of size $k$) of its children sectors. Evaluating the binary tree bottom-up depth-wise is possible, but is also costly since the leaf layer has $n$ sectors leading to $O(nk)$ cost for storing the output values of children sectors. Rather, we can proceed greedily, evaluating leaf sectors left-to-right and propagating values up a sectors as soon as all of its children sectors have been evaluated. A simple recursive argument shows that this procedure requires storing only $O(k \log n)$ intermediate values at a time, and is provably optimal.

**Proposition 1.** *Evaluating a complete binary-tree computation graph requires $\Theta(\log n)$ memory, where $n$ is the number of leaves in the complete binary-tree, and the memory cost of storing each intermediate value and performing each computation is assumed to be a constant.*

*Proof.* Perform the sequence of computations based on a post-order traversal of the binary tree, erasing children values from memory when their parent value has been computed. This process requires storing at most $\log n + 1$ intermediate values at any time. To show optimality, let $C(i)$ denote the optimal memory cost needed to compute the value for a node $i$. For non-leaf nodes $i$, Let $j_1$ be the first child of $i$ that we begin to process, and $j_2$ be the second. Since we processed $j_1$ first (which requires at least 1 unit of memory throughout), we have $C(i) \geq 1 + C(j_2)$. As the base case, for leaf nodes $i$, we have $C(i) = 1$. Thus, the optimal memory cost for the whole computation is at least $C(\text{root}) = \log n + 1$. $\square$

**Corollary 1.** *Evaluating a HyperSPN with an underlying RAT-SPN structure requires $O(k \log n + h(k^2 + rn))$ memory.*

The cost in Corollary 1 can be broken down into two parts, where $O(k \log n)$ memory is required for storing intermediate values in the binary-tree computation graph, and $O(h(rn + k^2))$ memory is required for storing the HyperSPN parameters and materializing the SPN one sector at a time.

The takeaway is that both storing and evaluating a HyperSPN require less memory than storing and evaluating the underlying SPN, which require $O(rnk^2)$ memory. This memory efficiency arises from our key insight of using sector abstractions to enable 1) encoding the SPN parameters as embeddings, 2) forgoing storage of the nodes and edges in the SPN network, and 3) mapping the SPN to a binary-tree computation graph that allows for a low-memory evaluation order.

## 4  Experiments

We experiment with HyperSPN on three sets of density estimation tasks: the Twenty Datasets benchmark, the Amazon Baby Registries benchmark, and the Street View House Numbers (SVHN) dataset. All runs were done using a single GPU. We primarily compare with weight decay as the competing regularization method. We also include comparisons with smaller SPNs that have the same degrees of freedom as our HyperSPN, and with results reported from other works in literature. We omit results on dropout because in our experimentation (and as suggested by [25]), dropout on SPNs worked poorly for these generative density estimation tasks. In all three sets of tasks, we find that HyperSPNs give better generalization performance than weight decay, when comparing using the same underlying SPN structure and optimizer. Furthermore, HyperSPNs outperforms results reported in literature on EiNETs trained using the same optimization method (Adam), and is competitive against results reported on stochastic EM (sEM).

### 4.1  Twenty Datasets

The Twenty Datasets benchmark [12] is a standard benchmark used in comparing PC models. Our experiment aims to compare the use of HyperSPN against the baseline of regularization via weight decay, using the same underlying SPN structure for both. We use the RAT-SPN structure described in Section 2.1, choosing layer size parameter $k = 5$ and replicas $r = 50$, randomizing the variable orders for each replica. The number of learnable parameters in this SPN structure is shown in the **# Params** column under Weight Decay in Table 1. For Weight Decay, we vary the weight decay value between 1e-3, 1e-4, and 1e-5.

The HyperSPN uses the exact same underlying SPN structure, along with an external neural network (a 2 layer MLP of width 20) and embeddings of dimension ranging between $h = 5, 10, 20$. The number of learnable parameters in the HyperSPN is around 5 times smaller, as shown in the **# Params** column under HyperSPN. We describe more details of our training setup in the Appendix.

In Table 1, we show in bold the better of the results between Weight Decay and HyperSPN. We see that HyperSPN outperforms Weight Decay on all but two of the datasets. On top of that, storing and evaluating the HyperSPN can be much more memory-efficient, as shown by the reduced number of trainable parameters and as described by the techniques in Section 3.1.

Table 1: Twenty Datasets. We plot the test log-likelihood and the number of trainable parameters. Bold values indicate the best results between the two regularization strategies of Weight Decay and HyperSPN, using the same SPN structure and optimizer. Underlined values indicate the best results when compared also to reported values in literature using EiNETs/Adam⋆ and RAT-SPN/sEM†.

| Name | Variables | Adam Weight Decay | | Adam HyperSPN | | Adam ⋆ | sEM † |
| | | Log-LH | # Params | Log-LH | # Params | Log-LH | Log-LH |
|---|---|---|---|---|---|---|---|
| NLTCS | 16 | -6.02 | 40050 | **_-6.01_** | 9115 | -6.04 | _-6.01_ |
| MSNBC | 17 | **-6.04** | 42550 | -6.05 | 9615 | _-6.03_ | -6.04 |
| KDDCup2k | 64 | -2.14 | 160050 | **_-2.13_** | 33115 | -2.15 | _-2.13_ |
| Plants | 69 | -13.41 | 172550 | **_-13.27_** | 35615 | -13.74 | -13.44 |
| Audio | 100 | -40.14 | 250050 | **_-39.74_** | 51115 | -40.22 | -39.96 |
| Jester | 100 | -52.99 | 250050 | **_-52.74_** | 51115 | -53.10 | -52.97 |
| Netflix | 100 | -57.18 | 250050 | **_-56.62_** | 51115 | -57.10 | -56.85 |
| Accidents | 111 | -35.55 | 277550 | **_-35.40_** | 56615 | -37.45 | -35.49 |
| Retail | 135 | -10.90 | 337550 | **_-10.89_** | 68615 | -10.97 | -10.91 |
| Pumsb-star | 163 | -31.08 | 407550 | **_-31.07_** | 82615 | -39.23 | -32.53 |
| DNA | 180 | **-98.42** | 450050 | -98.79 | 91115 | -97.68 | _-97.23_ |
| Kosarek | 190 | **_-10.88_** | 475050 | -10.90 | 96115 | -10.92 | -10.89 |
| MSWeb | 294 | -10.14 | 735050 | **_-9.90_** | 148115 | -10.26 | -10.12 |
| Book | 500 | **-34.84** | 1250050 | -34.86 | 251115 | -35.15 | _-34.68_ |
| EachMovie | 500 | -52.85 | 1250050 | **_-51.62_** | 251115 | -55.49 | -53.63 |
| WebKB | 839 | -159.68 | 2097550 | **-157.69** | 420615 | -160.51 | _-157.53_ |
| Reuters-52 | 889 | -90.15 | 2222550 | **_-86.12_** | 445615 | -92.76 | -87.37 |
| 20Newsgrp | 910 | -154.36 | 2275050 | **-152.49** | 456115 | -154.41 | _-152.06_ |
| BBC | 1058 | -262.77 | 2645050 | **-254.44** | 530115 | -267.86 | _-252.14_ |
| Ad | 1556 | -54.82 | 3890050 | **_-28.58_** | 779115 | -63.82 | -48.47 |

On the two right-most columns, we directly copy over results reported in literature. We compare with training EiNETs on Adam (the same optimizer as our setup) [24], and training RAT-SPNs on sEM (a similar network structure as our setup) [25]. We see that HyperSPN also compares favorably to these reported results (the best values underlined).

To illustrate the regularization effects of HyperSPN, we plot training curves on the training and validation data for the Plants and for Pumsb-star datasets. We use the same HyperSPN model described above. For clarity, let $M$ denote the size of its underlying SPN and $m$ denote its true number of learnable parameters (those in its external neural network and embeddings). We then construct two SPNs – one with size $M$ (SPN-Large) and one with size $m$ (SPN-Small) – and overlay their training curves in Figure 5. For both plots in Figure 5a and 5b, SPN-Large suffers from overfitting on the training data (presumably due to the large degrees of freedom), and SPN-Small fits the data poorly (presumably limited by the size of the SPN). Instead, HyperSPN strikes the balance between the expressiveness of a large underlying SPN, and regularization properties from the compact neural network with low degrees of freedom. This translates to better generalization on the validation data shown in the plots of Figure 5, and on the testing data shown in Table 1.

## 4.2 Amazon Baby Registries

We conduct similar experiments on the Amazon Baby Registries dataset [8]. We repeat the same experimental setup of maximizing the log-likelihood of data, and use the same model structure for HyperSPN and Weight Decay as their respective structures described in the previous section.

In Table 2, we see that HyperSPN clearly outperforms Weight Decay as a regularization technique, giving better test log-likelihoods (in bold) while using around 5 times fewer trainable parameters. On the right-most column we include results as reported in literature [37] on training EiNETs using sEM. Although the comparison with this right-most column is hard to interpret due to the differences in model architecture, optimization procedure, and regularization, we still note that HyperSPNs compare favorably on the majority of the datasets (best values underlined).

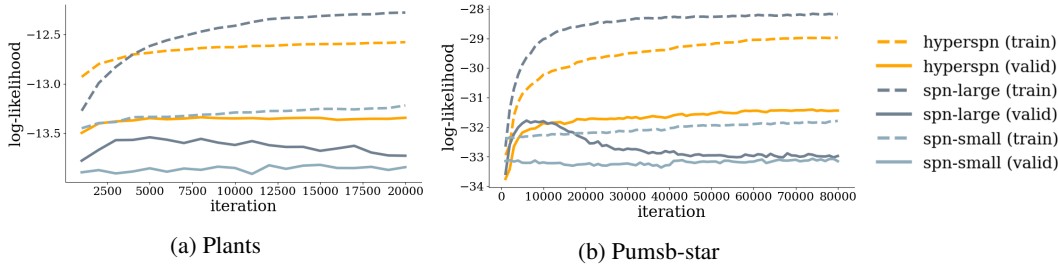

(a) Plants                (b) Pumsb-star

Figure 5: We plot the training curves on Plants and Pumsb-star from the Twenty Datasets benchmark. The HyperSPN has the same underlying SPN as SPN-Large, but limits its degrees of freedom to the same as those of SPN-Small. In both plots, HyperSPN fits the data better than SPN-Small, and is better regularized than SPN-Large, leading to the best results on the validation set (solid orange line).

The experimental results on the Amazon Baby Registries benchmark confirm our findings in the previous subsection. On both tasks, HyperSPNs generalizes much better than Weight Decay when compared on the same underlying SPN structure, while enabling more memory-efficient storage and evaluation of the model.

Table 2: Amazon Baby Registries. We plot the test log-likelihood and the number of trainable parameters. Bold values indicate the best results between the two regularization strategies of Weight Decay and HyperSPN, using the same SPN structure and optimizer. Underlined values indicate the best results when compared also to reported values in literature using EiNETs/sEM*.

| Name | Variables | Adam Weight Decay | | Adam HyperSPN | | sEM * |
|---|---|---|---|---|---|---|
| | | Log-LH | # Params | Log-LH | # Params | Log-LH |
| Apparel | 100 | -9.32 | 250050 | **-9.28** | 50655 | -9.24 |
| Bath | 100 | -8.54 | 250050 | **-8.52** | 50655 | -8.49 |
| Bedding | 100 | -8.59 | 250050 | **-8.57** | 50655 | -8.55 |
| Carseats | 34 | -4.72 | 85050 | **-4.65** | 17655 | -4.72 |
| Diaper | 100 | -9.99 | 250050 | **-9.91** | 50655 | -9.86 |
| Feeding | 100 | -11.35 | 250050 | **-11.30** | 50655 | -11.27 |
| Furniture | 32 | -4.54 | 80050 | **-4.32** | 16655 | -4.38 |
| Gear | 100 | **-9.21** | 250050 | **-9.21** | 50655 | -9.18 |
| Gifts | 16 | -3.43 | 40050 | **-3.40** | 8655 | -3.42 |
| Health | 62 | -7.49 | 155050 | **-7.41** | 31655 | -7.47 |
| Media | 58 | -7.86 | 145050 | **-7.83** | 29655 | -7.82 |
| Moms | 16 | -3.48 | 40050 | **-3.47** | 8655 | -3.48 |
| Safety | 36 | -4.48 | 90050 | **-4.34** | 18655 | -4.39 |
| Strollers | 40 | -5.25 | 100050 | **-5.00** | 20655 | -5.07 |
| Toys | 62 | -7.83 | 155050 | **-7.79** | 31655 | -7.84 |

**Sample Quality**     In addition to generalization performance on log-likelihood, we can examine the sample quality of HyperSPNs in comparison to standard SPNs. To estimate sample quality, we take the approach of applying kernel density estimation / Parzen windows [3] using the test dataset. Out of the 35 datasets from the above Twenty Datasets and Amazon Baby Registries, we observe that HyperSPNs report better sample quality for 29 of them, compared to SPNs trained with weight decay. We report the full results of this experiment in the Appendix.

### 4.3 Street View House Numbers (SVHN)

Lastly, we experiment on the SVHN dataset [20]. We use the infrastructure provided by [24], which groups the train/valid/test data into 100 clusters, then learns a separate PC for each cluster. The final density estimator is a combination of all the individual PCs under one mixture distribution. Here, we

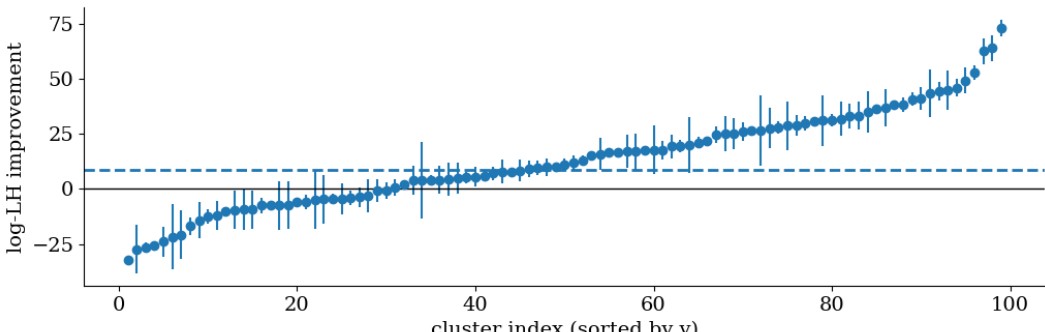

Figure 6: We plot the improvement in test log-likelihood on the Street View House Numbers (SVHN) dataset when using HyperSPN instead of weight decay. Following the setup from [24], the dataset is split into 100 clusters and a separate HyperSPN is learned for each cluster. We sort the cluster index by improvement for readability. The dashed blue line corresponds to the average (weighted by cluster size) log-likelihood improvement of 9.

study the performance of each individual PC on each cluster separately, comparing between weight decay and HyperSPN when trained using Adam.

The underlying PC structure is an EiNET with leaves that hold continuous distributions [24]. To introduce soft weight-sharing of the HyperSPN on the EiNET, we partition the parameters into sectors of size $40$, which is the number of sum nodes used in each layer in the setup of [24]. We use an embedding size of $h = 8$, which reduces the number of learnable edge-parameters in the EiNET by about $5$ fold. In Figure 6, we plot the difference between HyperSPN and weight decay in log-likelihood on the test set, sorted in increasing order for readability. The higher the value, the more improvement we observe when using HyperSPN as the regularization method. In around $70\%$ of the clusters, we see an improvement in generalization, and on average (weighted by cluster size) the HyperSPN achieves an improvement in log-likelihood of $9 \pm 4$.

Our experiments suggest that HyperSPNs give a superior regularization strategy for PCs compared to the primary alternative of weight decay. HyperSPNs also lead to better performance than other reported approaches in literature that use the same optimizer (Adam), as well as other reported methods that use sEM. Our experimental results are general, showing that HyperSPNs are applicable across three standard benchmarks, across PCs with discrete and continuous leaf distributions, and across different state-of-the-art structures such as RAT-SPNs and EiNETs.

## 5 Related Work

Many existing works focus on structure learning for probabilistic circuits, covering a wide range of PC types [7, 28, 27, 15]. These approaches often regularize by limiting the complexity of the structure [7, 35]. However, structure learning in general can be tedious and difficult to tune [25].

Recent works have embraced a more deep-learning-like framework by prescribing the structure of the PC and learning only the model parameters [25, 24]. This framework leads to the training of large PCs with millions of parameters. However, little exploration has been done around regularization in this new parameter-centric framework. For generative settings, dropout was suggested as unsuitable [25], leaving only weight decay as the primary parameter regularization method. Concurrently, other works [16] look into data softening and entropy regularization techniques. In our work, we propose the new regularization strategy of HyperSPNs which show promising improvements in generalization.

Finally, the idea of generating parameters with an external neural network comes from HyperNetworks [11], which were introduced as an effective architecture for recurrent models. Soft weight-sharing strategies in general [21] have also shown good empirical results [2] and proven useful for neural network compression [32]. There have also been recent work on using similar techniques to parameterize conditional SPNs [29, 36].

# 6 Conclusion

We propose HyperSPNs, a new paradigm of training large probabilistic circuits using a small-scale external neural network to generate the parameters of the probabilistic circuit. HyperSPNs provide regularization by aggressively reducing the degrees of freedom through a soft weight-sharing strategy. We show empirical improvements in the generalization performance of PCs compared to competing regularization methods, across both discrete and continuous density estimation tasks, and across different state-of-the-art PC structures. As a bonus, HyperSPNs enable memory-efficient storage and evaluation, combining the expressiveness of large PCs with the compactness of small models.

**Limitations and Future Work**   Our analysis of the memory efficiency of HyperSPNs is limited their storage and evaluation, since training HyperSPNs involves manipulating of the gradients through the underlying SPN. Future work can look into combining HyperSPNs with techniques from memory-efficient back propagation [4, 10] to train very large SPNs that do not fit in memory.

# 7 Acknowledgments

This research was supported by NSF (#1651565, #1522054, #1733686), ONR (N00014-19-1-2145), AFOSR (FA9550-19-1-0024), ARO (W911NF-21-1-0125) and Sloan Fellowship.

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
