The code for our experiments is available at `https://github.com/AndyShih12/HyperSPN`.

## A   Hand-Crafted Example

To examine the merits of HyperSPNs as discussed in Section 3, we construct a hand-crafted dataset to test the three types of models described in Figure 4: SPN-Large, SPN-Small, and HyperSPN. The hand-crafted dataset is procedurally generated with 256 binary variables and 10000 instances, broken into train/valid/test splits at $70/10/20\%$. The generation procedure is designed such that the correlation between variable $i$ and $j$ is dependent on the path length between leaves $i$ and $j$ of a complete binary tree over the 256 variables. The exact details can be found in our code.

SPN-Large has the same number of SPN edges as the HyperSPN, while SPN-Small has roughly the same number of trainable parameters as HyperSPN. Both SPN-Large and SPN-Small are regularized via weight-decay. As we can see in Table 3, HyperSPN gives the best generalization performance on the test split of our hand-crafted dataset, when compared to standard SPNs with either similar number of SPN edges (SPN-Large) and similar number of trainable parameters (SPN-Small).

Table 3: Testing HyperSPNs on a hand-crafted toy dataset with 256 variables.

|           | Log-Likelihood | # Params |
|-----------|----------------|----------|
| SPN-Large | -166.90 ± 0.03 | 640050   |
| SPN-Small | -167.00 ± 0.01 | 102450   |
| HyperSPN  | **-166.32 ± 0.04** | 129115 |

## B   Experimental Details

Here, we provide more details on our experimental setup. In Table 4, we give the hyperparameters used for training our models on the Twenty Datasets and Amazon Baby Registries benchmarks. For both methods we do early stopping by training until the validation performance plateaus/declines (we train some up to 80k steps). Then we take the version of the model that performed best on the validation set, and use it for evaluation on the test set.

Recall that for the standard SPN, we take gradient descent steps on the mixture weights of the SPN. For the HyperSPN, we take gradient descent steps on the parameters of the external neural network. Empirically, we found that higher learning rates are more suitable for the SPN (e.g. 2e-2), and lower learning rates are more suitable for the HyperSPN (e.g. 5e-3).

Table 4: Training hyperparameters for Twenty Datasets and Amazon Baby Registries

|               | SPN (Weight Decay) | HyperSPN     |
|---------------|--------------------|--------------|
| Learning Rate | 2e-2               | 5e-3         |
| Weight Decay  | {1e-3, 1e-4, 1e-5} | –            |
| Embedding Dim | –                  | {5, 10, 20}  |
| Batch Size    | 500                | 500          |

For the SVHN experiment, we build on the code provided at `https://github.com/cambridge-mlg/EinsumNetworks`. We train each method for 100 epochs using Adam, and also use early stopping based on the validation set. The weight decay hyperparameter used for the weights of the models are the same as that shown in Table 4, and we scale down the learning rate for both models to 1e-3 and 3e-4, respectively. We found the slower learning rate to be more suitable for both models on this benchmark, with HyperSPNs still giving the better performance.

For the Twenty Datasets and Amazon Baby Registries, the leaf nodes are binary indicator random variables, hence there are no trainable parameters. For SVHN, the leaf nodes are factorized Gaussians. The training of the leaf distributions was kept the same for SPNs and HyperSPNs (i.e. for HyperSPNs, we do not generate the parameters of the leaf distributions using the external neural network).

# C   Additional Experiments and Ablations

We provide additional experiments that examine different hyperparameter choices for the embedding size $h$ and for the weight decay parameter. We also show more detailed results for sample quality as measured using Parzen windows. Finally, we report error bars for the experiments from Tables 1 & 2.

## C.1   Weight Decay and Embedding Size $h$

We try out different hyperparameter values for the weight decay value used for regularizing standard SPNs, and for the embedding size of the HyperSPNs. We report these results in Table 5, where we see that weight decay of 1e-4 for standard SPNs and embedding size of $h = 10$ for HyperSPNs gave slightly better performance than the alternative settings.

Table 5: For standard SPNs (left), we vary the weight decay value between 1e-3, 1e-4, and 1e-5. For HyperSPNs (right), we vary the embedding size $h$ between 5, 10, and 20.

| Name | Variables | SPN | | | HyperSPN | | |
|---|---|---|---|---|---|---|---|
| | | $w =$1e-3 | $w =$1e-4 | $w =$1e-5 | $h = 5$ | $h = 10$ | $h = 20$ |
| NLTCS | 16 | -6.02 | -6.02 | -6.02 | -6.03 | -6.01 | -6.02 |
| MSNBC | 17 | -6.06 | -6.04 | -6.04 | -6.05 | -6.06 | -6.07 |
| KDDCup2k | 64 | -2.14 | -2.14 | -2.14 | -2.14 | -2.13 | -2.13 |
| Plants | 69 | -13.44 | -13.41 | -13.45 | -13.31 | -13.27 | -13.27 |
| Audio | 100 | -40.16 | -40.14 | -40.17 | -39.86 | -39.74 | -39.75 |
| Jester | 100 | -53.01 | -52.99 | -53.05 | -52.92 | -52.74 | -52.83 |
| Netflix | 100 | -57.18 | -57.20 | -57.21 | -56.73 | -56.66 | -56.62 |
| Accidents | 111 | -35.94 | -35.55 | -35.65 | -36.02 | -35.52 | -35.40 |
| Retail | 135 | -10.92 | -10.92 | -10.90 | -10.89 | -10.92 | -10.92 |
| Pumsb-star | 163 | -31.89 | -31.08 | -31.48 | -31.46 | -31.39 | -31.07 |
| DNA | 180 | -98.59 | -98.42 | -98.45 | -98.79 | -99.05 | -98.88 |
| Kosarek | 190 | -10.91 | -10.89 | -10.88 | -10.90 | -10.92 | -10.92 |
| MSWeb | 294 | -10.28 | -10.14 | -10.14 | -9.93 | -9.92 | -9.90 |
| Book | 500 | -34.90 | -34.84 | -35.02 | -34.95 | -35.02 | -34.86 |
| EachMovie | 500 | -52.85 | -53.21 | -54.01 | -51.62 | -52.10 | -52.00 |
| WebKB | 839 | -159.68 | -160.10 | -160.06 | -157.69 | -158.35 | -158.24 |
| Reuters-52 | 889 | -92.82 | -90.15 | -90.99 | -86.93 | -86.12 | -86.76 |
| 20Newsgrp | 910 | -154.36 | -154.70 | -155.01 | -152.57 | -152.82 | -152.49 |
| BBC | 1058 | -267.47 | -262.77 | -268.99 | -256.07 | -254.44 | -255.81 |
| Ad | 1556 | -56.34 | -54.90 | -54.82 | -31.50 | -28.58 | -29.84 |
| Apparel | 100 | -9.32 | -9.33 | -9.33 | -9.30 | -9.28 | -9.28 |
| Bath | 100 | -8.54 | -8.59 | -8.60 | -8.52 | -8.52 | -8.52 |
| Bedding | 100 | -8.64 | -8.59 | -8.63 | -8.59 | -8.58 | -8.57 |
| Carseats | 34 | -4.72 | -4.82 | -4.76 | -4.77 | -4.65 | -4.66 |
| Diaper | 100 | -9.99 | -10.03 | -10.00 | -9.91 | -9.94 | -9.93 |
| Feeding | 100 | -11.37 | -11.35 | -11.41 | -11.31 | -11.30 | -11.31 |
| Furniture | 32 | -4.56 | -4.54 | -4.60 | -4.48 | -4.46 | -4.32 |
| Gear | 100 | -9.24 | -9.21 | -9.21 | -9.41 | -9.21 | -9.21 |
| Gifts | 16 | -3.48 | -3.43 | -3.48 | -3.44 | -3.40 | -3.42 |
| Health | 62 | -7.49 | -7.50 | -7.49 | -7.75 | -7.41 | -7.43 |
| Media | 58 | -7.86 | -7.91 | -7.90 | -7.87 | -7.83 | -7.87 |
| Moms | 16 | -3.49 | -3.48 | -3.49 | -3.48 | -3.50 | -3.47 |
| Safety | 36 | -4.48 | -4.48 | -4.57 | -4.44 | -4.37 | -4.34 |
| Strollers | 40 | -5.27 | -5.29 | -5.25 | -5.23 | -5.02 | -5.00 |
| Toys | 62 | -7.83 | -7.83 | -7.83 | -7.83 | -7.81 | -7.79 |
| Average | - | -37.18 | **-36.91** | -37.17 | -35.79 | **-35.63** | -35.68 |

## C.2    Sample Quality

Next, we report estimates of sample quality of the trained SPN / HyperSPN using Parzen windows on the test data. We treat the binary data as real vectors, and use a Gaussian kernel with a fixed variance on each of the test data points. Then, we sample $500$ data points from the trained SPN / HyperSPN, and compute the log-likelihood of the samples.

Table 6: We examine an estimate of sample quality using Parzen windows on the test data, taking $500$ samples per dataset. We see that HyperSPNs give better sample quality under this metric (higher is better).

| Name | Variables | SPN | HyperSPN |
|---|---|---|---|
| NLTCS | 16 | -2.346 | -2.348 |
| MSNBC | 17 | -2.247 | -2.247 |
| KDDCup2k | 64 | -2.066 | -2.066 |
| Plants | 69 | -2.927 | -2.926 |
| Audio | 100 | -3.606 | -3.594 |
| Jester | 100 | -4.329 | -4.319 |
| Netflix | 100 | -4.470 | -4.467 |
| Accidents | 111 | -3.557 | -3.554 |
| Retail | 135 | -2.290 | -2.290 |
| Pumsb-star | 163 | -4.247 | -4.238 |
| DNA | 180 | -5.693 | -5.676 |
| Kosarek | 190 | -2.310 | -2.318 |
| MSWeb | 294 | -2.299 | -2.299 |
| Book | 500 | -2.758 | -2.740 |
| EachMovie | 500 | -4.053 | -3.972 |
| WebKB | 839 | -6.307 | -6.111 |
| Reuters-52 | 889 | -4.632 | -4.558 |
| 20Newsgrp | 910 | -5.683 | -5.705 |
| BBC | 1058 | -9.401 | -8.942 |
| Ad | 1556 | -3.187 | -3.147 |
| Apparel | 100 | -2.257 | -2.251 |
| Bath | 100 | -2.219 | -2.209 |
| Bedding | 100 | -2.233 | -2.224 |
| Carseats | 34 | -2.151 | -2.152 |
| Diaper | 100 | -2.273 | -2.262 |
| Feeding | 100 | -2.308 | -2.301 |
| Furniture | 32 | -2.146 | -2.144 |
| Gear | 100 | -2.238 | -2.228 |
| Gifts | 16 | -2.134 | -2.134 |
| Health | 62 | -2.205 | -2.205 |
| Media | 58 | -2.229 | -2.227 |
| Moms | 16 | -2.135 | -2.135 |
| Safety | 36 | -2.141 | -2.142 |
| Strollers | 40 | -2.163 | -2.160 |
| Toys | 62 | -2.219 | -2.216 |
| Average | - | -3.185 | **-3.157** |

## C.3 Error Bars

Lastly, we report the main results from Table 1 and Table 2 with error bars, computed over 3 separate runs with different random seeds.

Table 7: We report the values from in Table 1 and Table 2 with error bars, computed over 3 separate runs. We denote statistical significance with a $\star$.

| Name | Variables | SPN | HyperSPN |
|------|-----------|-----|----------|
| NLTCS | 16 | -6.02 $\pm$ 0.00 | -6.01$^\star$ $\pm$ 0.00 |
| MSNBC | 17 | -6.04 $\pm$ 0.00 | -6.05 $\pm$ 0.01 |
| KDDCup2k | 64 | -2.14 $\pm$ 0.00 | -2.13$^\star$ $\pm$ 0.00 |
| Plants | 69 | -13.41 $\pm$ 0.03 | -13.27$^\star$ $\pm$ 0.06 |
| Audio | 100 | -40.14 $\pm$ 0.01 | -39.74$^\star$ $\pm$ 0.03 |
| Jester | 100 | -52.99 $\pm$ 0.02 | -52.74$^\star$ $\pm$ 0.02 |
| Netflix | 100 | -57.18 $\pm$ 0.01 | -56.62$^\star$ $\pm$ 0.01 |
| Accidents | 111 | -35.55 $\pm$ 0.07 | -35.40 $\pm$ 0.11 |
| Retail | 135 | -10.90 $\pm$ 0.01 | -10.89 $\pm$ 0.03 |
| Pumsb-star | 163 | -31.08 $\pm$ 0.12 | -31.07 $\pm$ 0.02 |
| DNA | 180 | -98.42$^\star$ $\pm$ 0.05 | -98.79 $\pm$ 0.09 |
| Kosarek | 190 | -10.88 $\pm$ 0.01 | -10.90 $\pm$ 0.03 |
| MSWeb | 294 | -10.14 $\pm$ 0.02 | -9.90$^\star$ $\pm$ 0.02 |
| Book | 500 | -34.84 $\pm$ 0.03 | -34.86 $\pm$ 0.02 |
| EachMovie | 500 | -52.85 $\pm$ 0.11 | -51.62$^\star$ $\pm$ 0.22 |
| WebKB | 839 | -159.68 $\pm$ 0.21 | -157.69$^\star$ $\pm$ 0.63 |
| Reuters-52 | 889 | -90.15 $\pm$ 0.49 | -86.12$^\star$ $\pm$ 0.13 |
| 20Newsgrp | 910 | -154.36 $\pm$ 0.04 | -152.49$^\star$ $\pm$ 0.67 |
| BBC | 1058 | -262.77 $\pm$ 0.10 | -254.44$^\star$ $\pm$ 0.29 |
| Ad | 1556 | -54.82 $\pm$ 0.97 | -28.58$^\star$ $\pm$ 0.20 |
| Apparel | 100 | -9.32 $\pm$ 0.01 | -9.28 $\pm$ 0.02 |
| Bath | 100 | -8.54 $\pm$ 0.01 | -8.52 $\pm$ 0.02 |
| Bedding | 100 | -8.59 $\pm$ 0.01 | -8.57$^\star$ $\pm$ 0.00 |
| Carseats | 34 | -4.72 $\pm$ 0.01 | -4.65$^\star$ $\pm$ 0.01 |
| Diaper | 100 | -9.99 $\pm$ 0.07 | -9.91$^\star$ $\pm$ 0.02 |
| Feeding | 100 | -11.35 $\pm$ 0.01 | -11.30$^\star$ $\pm$ 0.00 |
| Furniture | 32 | -4.54 $\pm$ 0.02 | -4.32 $\pm$ 0.13 |
| Gear | 100 | -9.21 $\pm$ 0.01 | -9.21 $\pm$ 0.01 |
| Gifts | 16 | -3.43 $\pm$ 0.02 | -3.40$^\star$ $\pm$ 0.00 |
| Health | 62 | -7.49 $\pm$ 0.01 | -7.41$^\star$ $\pm$ 0.02 |
| Media | 58 | -7.86 $\pm$ 0.01 | -7.83 $\pm$ 0.03 |
| Moms | 16 | -3.48 $\pm$ 0.00 | -3.47 $\pm$ 0.03 |
| Safety | 36 | -4.48 $\pm$ 0.01 | -4.34$^\star$ $\pm$ 0.01 |
| Strollers | 40 | -5.25 $\pm$ 0.01 | -5.00$^\star$ $\pm$ 0.03 |
| Toys | 62 | -7.83 $\pm$ 0.02 | -7.79 $\pm$ 0.03 |
| Average | - | -36.87 $\pm$ 0.07 | **-35.55$^\star$ $\pm$ 0.08** |