# OpenReview forum: "HyperSPNs: Compact and Expressive Probabilistic Circuits"
_NeurIPS.cc/2021/Conference — NeurIPS 2021 Poster_

### Official Review · Reviewer_ohxd · 2021-07-14

**Rating:** 7
**Confidence:** 4

**Summary:**

The paper proposes a novel method for regularizing large scale probabilistic circuits: The PC's parameters are partitioned into evenly sized sectors, and each sector is represented via a low dimensional embedding. A shared neural network is used to map the embeddings in to the PC's original parameter space. Experimental results show that this reduction in the number of trainable parameters yields improved generalization performance when compared to previous methods, mainly weight decay.

**Limitations And Societal Impact:**

The limitations and societal impact are adequately discussed.

**Main Review:**

### Originality / Significance
The proposed method is novel to my knowledge. It addresses a relevant problem, as the fast growth in the number of parameters is one of the main issues preventing RAT-SPNs and EiNets from scaling further.

The proposed method is intuitively plausible, and the theoretical analysis is correct.

### Evaluation
The proposed method is evaluated on a large variety of datasets, and the results are overall convincing.

Given that weight decay is the main point of comparison, it would be good run a hyperparameter search over its value, to verify that the chosen setting is sensible (and ideally optimal).

The robustness of the results in Tables 1 and 2 could also be improved by running the experiments multiple times and testing for significance.

For the SVHN experiment, it would be good to report the change in performance of the overall model, especially given the fact that the clusters are not uniform in size (i.e., given the reported data, it is not clear whether overall performance went up or down).

### Clarity
The paper is very well written and easy to follow.

### Summary
Overall, this paper makes a simple but useful contribution that is well presented, theoretically analyzed, and empirically evaluated.
I have therefore set my rating to accept, with the expectation that the authors will address the above comments on the empirical evaluation during the response phase.


**Time Spent Reviewing:**

4

---

> ### Author Response · Authors · 2021-08-10
> **Response**
>
> **run a hyperparameter search over its value**
>
> Thanks for the suggestion. We had picked the weight decay value based on ones reported in literature [33]. Based on your suggestion we ran a sweep over different values for weight decay (1e-2, 1e-3, 1e-4, 1e-5). We observed that 1e-4 gave the best results, so we will update the tables accordingly.
>
> | Weight Decay | Log-Likelihood (averaged over all datasets) |
> | :---: | :---: |
> | 1e-2 | -37.73 |
> | 1e-3 | -37.19 |
> | 1e-4 | -36.91 |
> | 1e-5 | -37.17 |
>
> | Method | Log-Likelihood (averaged over all datasets) |
> | :---: | :---: |
> | HyperSPN | -35.63 |
>
> For comparison HyperSPN still outperforms Weight Decay=1e-4 by a good margin (and on 31/35 datasets).
>
> **running the experiments multiple times and testing for significance**
>
> Thanks for the suggestion. We re-ran our experiments 3 times. Our results are significant for 27 out of the 31 datasets on which HyperSPN outperforms standard SPN. We will include more runs in the final version of the paper.
>
> **report the change in performance of the overall model, especially given the fact that the clusters are not uniform in size**
>
> This is a good point. We have computed the change in overall model, and we also re-ran the experiments more times as well. We observe that HyperSPNs give a log-likelihood improvement of 9 on the overall model (and is statistically significant).

---

> > ### Comment · Reviewer_ohxd · 2021-08-31
> > **Concerns Addressed**
> >
> > Thank you for the additional results. They have addressed my concerns, and I am therefore leaving my rating at 7.

---

### Official Review · Reviewer_BMBc · 2021-07-16

**Rating:** 5
**Confidence:** 4

**Summary:**

This work proposes a deep learning architecture, called HyperSPN, with built-in regularization. The architecture comprises two components: a probabilistic circuit (PC) and a neural network (NN). The main idea here is that the NN generates the parameters of the PC. The NN structure provides the regularization technique: parts of the NN embeddings are shared when generating the PC parameters.
The paper's theoretical contribution is the architecture involving the PC and NN for efficient training and inference tasks. Practical contributions include experiments in three different problem domains.

**Limitations And Societal Impact:**

The paper briefly discusses one limitation in the Conclusion section. Future works include combining HyperSPNs with techniques from memory-efficient back-propagation.


**Main Review:**

The originality of this work is not clear from the manuscript. The idea of conditioning a PC with weights from a NN has been explored in the PC literature. Some examples are [1] and, more recently, [2]. This paper can benefit from including a proper discussion on how the proposed method differs from those published. It would be interesting to discuss also how it compares to not recent but similar techniques, such as in [3].
The idea of regularizing the PC by sharing some of the NN architecture is smart and with promising results.

The paper's theoretical contributions are sound, to the best of my knowledge. There are only two derivations, namely Proposition 1 and Corollary 1, and both are well presented.

The manuscript is well written, and it has a good presentation. The architecture's description is supported by informative Figures 1, 2, and 3, and the experiments are also well described. This description is essential for the work's reproducibility.

The motivation of the manuscript is significant for the research community. Large models are prominent now in the literature and, thus, efficient regularization techniques are welcome. However, the manuscript should clarify this work's main contributions and how the work adds to previous similar papers. This clarification will strengthen this work's significance.
Another suggestion for improving significance is to show results on specific tasks. Improvements on log-likelihood, at times, do not reflect on improvements of particular tasks, such as classification or sampling. It would be interesting to see results on such tasks, besides the log-likelihood numbers.

Reference
* [1] "Conditional Sum-Product Networks: Imposing Structure on Deep Probabilistic Architectures." PDF: https://arxiv.org/abs/1905.08550
* [2] "RECOWNs: Probabilistic Circuits for Trustworthy Time Series Forecasting." PDF: https://arxiv.org/abs/2106.04148
* [3] "Hierarchical mixtures of experts and the EM algorithm." PDF: https://www.cs.toronto.edu/~hinton/absps/hme.pdf


**Time Spent Reviewing:**

2

---

> ### Author Response · Authors · 2021-08-10
> **Response**
>
> **The originality of this work is not clear from the manuscript...This paper can benefit from including a proper discussion on how the proposed method differs from those published.**
>
> Thank you for these references. These works were an interesting read, but rather different from our problem formulation and approach. (We also noticed that [2] was posted after our submission deadline.) Nonetheless we would be happy to discuss these in the final version of the paper.
>
> - [3]: These are mixture models arranged in a tree structure. They have no factorization, making them a limited form of SPNs without product nodes. Since these models do not compose together leaf nodes via products, they require the use of complex leaf distributions, making exact marginals intractable in general.
> - [1,2]: These works focus on a different problem setup: they study discriminative settings where the output is high-dimensional and structured. They do use a neural network, but use it to map the inputs X to a multi-dimensional label space Y,  for which they model using an SPN.
>
> 	There are key differences are in architecture, problem setting, and technique:
> 	- Architecture: Their design is to learn “hundreds” of SPNs at the same time. We are only learning one SPN.
> 	- Problem Setting: They operate in the discriminative setting; we operate in the generative setting. Namely, learning a separate SPN for each input (as they did) would not give us a valid probability distribution over the inputs, which is what we need under our density estimation setting.
> 	- Technique: They run a separate clustering phase (k-means) and use the neural network as gating-functions to predict the one-hot cluster assignments of inputs (more akin to a classification setup). This is different from our use of the neural network to predict weights from embeddings.
>
> Again, we will include these discussions in the updated version of the paper.
>
> **should clarify this work's main contributions and how the work adds to previous similar papers**
>
> Our main contribution is, as Reviewer ohxd put it, “a novel method for regularizing large scale probabilistic circuits”. As discussed above, our problem setting and approach are different from the references provided. We focus on large SPNs with prescribed structure (with no structure learning phase) and improve upon the SOTA architectures (RATSPNs, EiNETs) by proposing a more effective regularization technique.
>
> **show results on specific tasks... such as classification or sampling**
>
> Thanks for the suggestion. Since we focus on the generative setting, measuring sample quality is indeed a good idea. We take the approach of measuring sample quality via kernel density estimation / Parzen windows [4] on the dataset. Under this metric, we also observe increased sample quality when using HyperSPNs.
>
> For each dataset we took 500 samples from the final SPN, and computed the average sample quality. We observed improved sample quality in 29/35 datasets (significant on 19 out of the 29). To avoid clutter, here we report the sample quality, averaged over all the datasets. We will include the full results for each dataset separately in the final version of the paper.
>
> | Method | Sample Quality (higher is better) |
> | :---: | :---: |
> | HyperSPN | -3.15 |
> | SPN (small/large) | -3.18 |
>
> [4] Bengio, Y., Mesnil, G., Dauphin, Y., and Rifai, S. Better mixing via deep representations.

---

> > ### Author Response · Authors · 2021-08-31
> > **Follow Up**
> >
> > Please let us know if you would like further discussion around the provided references. We hope we have clarified the originality of our work. We have also incorporated your suggestion on examining sample quality. We hope you will reconsider your score in light of the response.

---

### Official Review · Reviewer_UiY3 · 2021-07-18

**Rating:** 4
**Confidence:** 5

**Summary:**

The paper presents HyperSPN, a way to reguralize the learning of Sum-Product networks (SPNs).

**Limitations And Societal Impact:**

The main limitation regrds the clarity of the proposed method

**Main Review:**

One advantage of the proposed approach is its memory efficiency.

The main problem of the paper regards the clarity of the proposed approach.
- It is not clear how the embeddings are defined. The relation between the embeddings and the SPN is not defined.
- The structure of the neural network used to learn the parameters is not defined
- The loss used to optimize the parameters is not reported
- How the embeddings are used by the neural network is not discussed

The experimental results prove an improvement of the proposed method when compared to weight decay. However, state of the art results on Twenty Datasets are not reported for a comparison.

Concluding, the idea seems to be interesting but the paper does not prove its validity. The approach should be better explained and the experiments should be extended.

**Time Spent Reviewing:**

3

---

> ### Author Response · Authors · 2021-08-10
> **Response**
>
> **It is not clear how the embeddings are defined. The relation between the embeddings and the SPN is not defined.**
>
> The embeddings are h-dimensional real-valued vectors -- we learn one embedding vector for each sector in the SPN. They are initialized randomly and learned (jointly with the neural network) via gradient descent. The embeddings can be viewed as low-dimensional encodings of the parameter weights for the SPN sectors, and the neural network can be viewed as the non-linear decoder that maps the embeddings to the SPN parameters.
>
> **The structure of the neural network used to learn the parameters is not defined**
>
> The structure of the neural network is described on Line 224. We use a MLP with two hidden-layers with width 20.
>
> **The loss used to optimize the parameters is not reported**
>
> The loss is the average negative log-likelihood on the training dataset, as is standard for density estimation tasks. We optimize the embeddings and the neural network via gradient descent. More details regarding the architecture and the optimization can be found in the Appendix and the supplementary code.
>
> **How the embeddings are used by the neural network is not discussed**
>
> The embeddings are the inputs to the neural network, and the corresponding outputs are used as the parameter weights of the SPN. We discuss this in Lines 137-148.
>
> **state of the art results on Twenty Datasets are not reported for a comparison**
>
> We do report state-of-the-art results on Twenty Datasets and Amazon Baby Registries for large SPNs with prescribed structure (i.e., no structure learning phase). These are the most relevant to our approach, since we are interested in pushing along the trend of learning large SPNs with no expensive structure learning phase.
>
> We did not include SOTA results from structure learning approaches. These are indeed better (for example, ID-SPNs), but are not scalable. Structure learning can take days of computation, whereas using prescribed structures, as we did, only takes minutes. We are happy to still include results from structure learning approaches if the reviewer wishes.
>
> **paper does not prove its validity**
>
> We respectfully disagree. Our experiments show generalization improvements across many datasets (Table1, Table2, Figure6), and we provide insight into this regularization effect in Figure5. We also support the idea of memory-efficient model evaluation with proofs.
>
> **The approach should be better explained and the experiments should be extended.**
>
> We hope we have clarified all the questions you brought up. Please let us know which parts can be better explained. Based on the suggestion of Reviewer BMBc we have additionally included metrics on sample quality using kernel density estimation. We agree with the other reviewers that the experiments are “quite comprehensive” and that the results are “promising” and “overall convincing”.

---

> > ### Author Response · Authors · 2021-08-31
> > **Follow Up**
> >
> > Please let us know if you would like any further clarifications regarding the proposed approach. Our earlier response pointed out the embedding definitions, the neural network structure, and the loss function. We hope you will reconsider your score in light of the response.

---

### Official Review · Reviewer_moxD · 2021-07-18

**Rating:** 7
**Confidence:** 4

**Summary:**

This paper presents a novel method to regularize very large probabilistic circuits, mainly large sum-product networks (SPNs). The idea is to train a small external neural network to generate the parameters (mixture weights) of a large SPN. The neural network itself is defined by a conservative number of parameters. Given the structure of an SPN, the parameter generation process first learns low dimensional embeddings that are representative of groups of parameters (called sectors) in the SPN. These embeddings are then fed into the external neural network which outputs the actual parameters for the SPN. Both the embeddings and the external neural network can be jointly trained using gradient descent. The neural network imposes soft weight-sharing over the SPN parameters which results in limiting the degrees of freedom of the model, therefore reducing potential overfitting. On a variety of benchmark datasets, both discrete and continuous, the authors empirically evaluated the proposed technique to regularize large SPNs and found that their method outperformed the well-known weight decay regularizer in terms of accuracy (log-likelihood scores) with significantly fewer parameters.


**Limitations And Societal Impact:**

yes

**Main Review:**

This paper makes an interesting novel contribution. The main goal is to reduce overfitting of large SPNs by limiting their degrees of freedom without limiting their expressivity. Learning smaller SPNs can reduce degrees of freedom at the cost of their expressivity.
Here the authors propose an alternative method where the degrees of freedom of the model is indirectly controlled by a small-scale external neural network which offers to impose a soft weight-sharing strategy over the parameters. This soft weight-sharing does not necessarily reduce the total number of parameters (clusters) of the model but restricts their structure and thus restrictis their degrees of freedom. The proposed technique is not only an effective regularizer but also an efficient means to compactly represent SPNs. One can easily opt to store only the neural network parameters and the embeddings which together have much fewer parameters than the underlying SPN itself. As the authors described this has the added benefit of reducing the memory footprint of SPNs during their evaluation as well.

The experiments seem quite comprehensive with a few exceptions. The authors tested their method on both discrete and continuous datasets and on two different structures (RAT-SPNs and EiNets). The results are promising. In majority of the cases the method reduced overfitting compared to the weight-decay strategy. The paper is written clearly and with quality.

Questions/Comments to authors:
1) An important piece of study that is missing is the effect of the length of 'h'. Two values were used in the experiments (5 and 8).
2) The values in tables 1 and 2 present average test log-likelihood?
3) The plot in figure 5 was quite informative. It will be useful if the scores for other datasets are also presented.


**Time Spent Reviewing:**

10

---

> ### Author Response · Authors · 2021-08-10
> **Response**
>
> **An important piece of study that is missing is the effect of the length of 'h'. Two values were used in the experiments (5 and 8).**
>
> Thanks for the suggestion. We re-ran our experiments, sweeping over values of $h=5,10,20$. The results were similar, with $h=10$ giving a slight edge. We will include the comparison in the appendix.
>
> For brevity, here are the test log-likelihoods, averaged over all the datasets. We will include the full results for each dataset separately in the final version of the paper.
>
> | h | Log-Likelihood (averaged over all datasets) |
> | :---: | :---: |
> | 5 | -35.79 |
> | 10 | -35.63 |
> | 20 | -35.68 |
>
> **The values in tables 1 and 2 present average test log-likelihood?**
>
> Yes, they present the average test log-likelihood.
>
> **The plot in figure 5 was quite informative. It will be useful if the scores for other datasets are also presented.**
>
> Thanks, we agree that they are informative! We will include more of these plots in the appendix.

---

### Author Response · Authors · 2021-08-10
**Response to All**

Thank you all for the helpful reviews. We appreciate that our paper is recognized as an “interesting novel contribution” that “addresses a relevant problem”. We agree that the experiments are “quite comprehensive” over a “large variety of datasets” and that the “results are overall convincing”. Finally, we are glad to hear that the paper is “written clearly and with quality”, and that our work is overall a “useful contribution that is well presented, theoretically analyzed, and empirically evaluated”.

We have addressed each of your concerns separately below. In summary, we have incorporated your combined suggestions of 1) trying different hyperparameters $h$ 2) trying different weight decay values 3) running multiple times for error bars, and 4) measuring sample quality (via kernel density estimation / Parzen windows). Thank you again for these suggestions, and we agree that they make the paper better.

---

### Decision · Program_Chairs · 2021-09-27

**Decision:**

Accept (Poster)

**Comment:**

Thank you for submitting your work to NeurIPS. The paper targets training large scale sum-product networks with a given structure. To this end, it pushes the idea of generating the mixture weights of
large PCs using a small-scale neural network. While there is a connection to Conditional SPNs as presented at PGM 2020, it actually covers the different aspect of learning SPNs and not conditional ones. It is very interesting to see that a shared neural network can be used during training to map the embeddings in to the PC's original parameter space. Scaling PCs (and not only conditional ones) is a really important question. Simple but highly effective and nice idea.